# Regret Bounds for Satisficing in Multi-Armed Bandit Problems

**Thomas Michel**                                                     *thomas.michel1@ens-paris-saclay.fr*
*Université Paris-Saclay*
*ENS Paris-Saclay*
*France*

**Hossein Hajiabolhassan**                                       *hossein.hajiabolhassan@medunigraz.at*
*Institute of Human Genetics*
*Diagnostic and Research Center for Molecular Biomedicine*
*Medical University of Graz*
*Austria*

**Ronald Ortner**                                                           *rortner@unileoben.ac.at*
*Lehrstuhl für Informationstechnologie*
*Montanuniversität Leoben*
*Austria*

**Reviewed on OpenReview:** *https://openreview.net/forum?id=QnT41ZGNh9*

## Abstract

This paper considers the objective of *satisficing* in multi-armed bandit problems. Instead of aiming to find an optimal arm, the learner is content with an arm whose reward is above a given satisfaction level. We provide algorithms and analysis for the realizable case when such a satisficing arm exists as well as for the general case when this may not be the case. Introducing the notion of *satisficing regret*, our main result shows that in the general case it is possible to obtain constant satisficing regret when there is a satisficing arm (thereby correcting a contrary claim in the literature), while standard logarithmic regret bounds can be re-established otherwise. Experiments illustrate that our algorithm is not only superior to standard algorithms in the satisficing setting, but also works well in the classic bandit setting.

## 1 Introduction

One of the reasons why reinforcement learning (RL) is difficult is that finding an *optimal* policy in general requires a lot of exploration. In practice however, we are often happy to perform a task just good enough. For example, when driving to work we will be content with a strategy that will let us arrive just in time, while the computation of a policy that is 'optimal' in some sense (e.g., along the shortest route, or as fast as possible) may be prohibitive. Accordingly, it is to be expected that when considering a *satisficing* objective aiming to find a solution that is above a certain satisfaction level it is possible to learn a respective policy much faster.

In this paper we investigate satisficing in the simplest RL setting, that is, the multi-armed bandit (MAB) problem. While computation is not an issue in the MAB setting, the goal of satisficing also makes sense for bandit applications in which one prefers a safe option that achieves a required level with high probability over exploration in order to find an optimal arm. This holds e.g. in many allocation problems for which bandit algorithms are a common choice. Often one is content with assigning a task to a resource (e.g., a machine or a person) that guarantees a certain level (e.g. concerning time or quality) for doing the task, while it may not pay off to risk more exploration to optimize with respect to the considered criterion. One such potential application is channel selection for a cognitive radio. Such a device is trying to select the best

available channel (e.g. having lowest utilization rate or highest reception quality) for which often multi-armed bandit algorithms are suggested (Kato et al., 2012; Avner & Mannor, 2014). However, it is reasonable to assume that choosing a channel that guarantees a certain reception quality is sufficient in this setting. Thus picking a channel that achieves the required level may be preferable to doing more exploration to identify the optimal channel.

In the following we introduce the notion of *satisficing regret* that measures the loss with respect to a given satisfaction level $S$. We first consider the realizable case, where this level can be achieved, that is, there is at least one arm whose expected reward is above the satisfaction level. In this setting, quite a simple algorithm can be shown to have constant satisficing regret (i.e., no dependence on the horizon $T$). For the general setting we provide the algorithm SAT-UCB which is able to extend this result, giving constant satisficing regret in the realizable case, while obtaining logarithmic bounds on the ordinary regret with respect to the optimal arm as for classic MAB algorithms such as UCB1 (Auer et al., 2002). Last but not least we confirm our theoretical findings in experiments which also show that SAT-UCB is competitive even in the standard setting. We also present a modification of SAT-UCB that experimentally performs even better, for which we were not able to show comparable theoretical guarantees however.

## 1.1  Setting

We consider the standard multi-armed bandit (MAB) setting with a set of $K$ arms, in the following denoted as $[\![1,K]\!] := \{1, 2, \ldots, K\}$. In discrete time steps $t = 1, 2, \ldots$ the learner picks an arm $A_t = i$ from $[\![1,K]\!]$ and observes a random reward $r_t$ drawn from a fixed reward distribution specific to the chosen arm $i$ with mean $\mu_i$. In the following we assume that the reward distributions for each arm are sub-Gaussian, that is, there is a constant $C > 0$ such that for all $\theta > 0$,

$$\mathbb{P}(|r_t| > \theta) \leqslant 2 \exp(-\tfrac{\theta^2}{C^2}).$$

Intuitively, this means that the tail bounds decay (at least) as fast as for a Gaussian distribution. Thus, all Gaussian distributions are sub-Gaussian. However, sub-Gaussian distributions also include other distributions that are commonly considered in the context of multi-armed bandit problems, such as Bernoulli distributions or generally any distribution with bounded support. Accordingly, in the following one can think of the reward distributions e.g. as bounded in some interval such as $[0, 1]$, which will contain the rewards $r_t$ as well as the means $\mu_i$ for each arm $i$. For alternative definitions and further details on sub-Gaussian distributions we refer to Section 2.5 of Vershynin (2018).

The usual performance measure for a learning algorithm in the MAB setting is the *(pseudo-)regret* after any finite number of $T$ steps, defined as

$$R_T := \sum_{t=1}^{T} \big( \mu_* - \mathbb{E}[\mu_{A_t}] \big),$$

where $\mu_* := \max_i \mu_i$ is the maximal mean reward over all arms.

In the satisficing setting however, we only care about whether an arm with mean reward $\geqslant S$ is chosen, where $S$ is a fixed level of satisfaction we aim at. Accordingly, we modify the classic notion of regret and consider what we call the *satisficing (pseudo-)regret* with respect to $S$ (short *$S$-regret*) defined as

$$R_T^S := \sum_{t=1}^{T} \max \big\{ S - \mathbb{E}[\mu_{A_t}], 0 \big\}.$$

This definition reflects that we are happy with any arm having mean reward $\geqslant S$ and that there is no benefit in overfulfilling the given satisfaction level $S$. Note that the $S$-regret will be linear in $T$ whenever there is no satisficing arm with mean reward $\geqslant S$, that is, if $\mu_* < S$. As will be discussed below, $S$-regret is closely related to the notion of *expected satisficing regret* as considered in a more general Bayesian setting introduced by Reverdy et al. (2017). We note that in the following we will consider algorithms which use randomization so that the expectation in the two regret definitions above is with respect to the stochasticity of the rewards as well as of the employed algorithm.

## 1.2 Related Work

While there are some connections to multi-criterion RL (Roijers et al., 2013), there is hardly any literature on satisficing in general RL settings. However, in the simple MAB setting satisficing and closely related ideas have been considered in some work, which we are going to discuss in the following.

Kohno & Takahashi (2017) and Tamatsukuri & Takahashi (2019) propose simple index policies for satisficing, which are experimentally evaluated. Tamatsukuri & Takahashi (2019) also show that the suggested algorithm converges to a satisficing arm and that the regret is finite if the satisfaction level is chosen to be between the reward of the best and the second-best arm.

Reverdy et al. (2017) consider a more general Bayesian setting, which also considers the learner's belief that some arm is satisficing. The notion of *expected satisficing regret* is introduced that measures the loss over all steps where a non-satisficing arm is chosen and the learner's degree of belief in the chosen arm was below some level $\delta \in [0, 1]$. For $\delta = 0$ our notion of $S$-regret as defined above is an upper bound on this expected satisficing regret. We note that in an earlier version (Reverdy et al., 2016) the expected satisficing regret has been defined slightly differently so that it coincides with our notion of $S$-regret when $\delta = 0$. Reverdy et al. (2017) present various bounds on the expected satisficing regret, including lower bounds as well as upper bounds for problems with Gaussian reward distributions when using adaptations of the UCL algorithm (Reverdy et al., 2014). We will discuss these bounds in more detail in Section 2 below.

Partly under the notion of *thresholding bandits* one can also find some work on satisficing in the pure-exploration setting. In this line of research the goal is to identify *all* arms above the satisfaction level, for which *sample complexity* or bounds on the *simple regret* are derived for a given confidence or a certain sample budget (Locatelli et al., 2016; Mukherjee et al., 2017; Kano et al., 2019). Unlike that, we consider *online* regret and we are content with choosing *any* arm above $S$. Note that any algorithm for pure exploration (Audibert et al., 2010) after any number of steps with high probability will identify an optimal or at least a satisficing arm. However, subsequent exploitation will always give linear regret due to the small but positive error probability so that a simple approach of first exploring and then exploiting does not work in general.

Related sample complexity bounds can also be found in (Mason et al., 2020) for identification of all $\varepsilon$-good arms. Closer to our setting is the problem of identifying an arbitrary arm among the top $m$ arms, for which sample complexity bounds are derived by Chaudhuri & Kalyanakrishnan (2017). A follow-up paper (Chaudhuri & Kalyanakrishnan, 2019) considers the sample complexity of the more general problem of identification of any $k$ of the best $m$ arms. None of these latter investigations however considers the online learning setting with regret as performance measure as we do.

A thresholding bandits contribution that is closer to our setting is that of Abernethy et al. (2016). Here the learner obtains a reward of 1 if the *actual* reward of the chosen arm (not the *expected* one as in our setting) is above a given threshold (which need not be fixed but may be different at any step), otherwise the reward is 0. The given regret bounds resemble those in the standard setting (i.e., are logarithmic in the horizon $T$) but with the mean rewards replaced by the so-called *survival functions* that specify the probability that some arm gives reward above a certain threshold. That is, unlike in our setting these bounds depend on the actual reward distributions and not just the mean rewards.

Another line of reseach that pursues similar ideas as our setting of satisficing is that of *conservative bandits*. Here the learner has an arm at her disposal that provides a baseline level (similar to our satisfaction level) one would not like to fall below, while trying to converge to an optimal arm. Thus Wu et al. (2016) present an algorithm that on the one hand with high probability stays above the baseline level at all time steps (with a certain amount of allowed error $\alpha$) and on the other hand has regret bounded similar to standard bandit algorithms (but with an additional dependence on $\alpha$).

Merlis & Mannor (2021) consider a related notion of so-called *lenient regret* that considers the loss with respect to $\mu_* - \varepsilon$ for a parameter $\varepsilon > 0$ that specifies the allowed deviation from the optimal mean reward $\mu_*$. The definition of lenient regret formally depends on a so-called $\varepsilon$-gap function. When choosing this function to be the hinge loss, lenient regret corresponds to $S$-regret when choosing $S := \mu^* - \varepsilon$. Merlis & Mannor (2021) show asymptotic upper bounds on the lenient regret for a version of Thompson sampling (Thompson,

1933) that match a given lower bound. Moreover, when $\mu_* > 1 - \varepsilon$ the lenient regret turns out to be constant. These results are discussed in more detail in Section 3 below.

Last but not least, Russo & Van Roy (2022) consider satisficing in a setting with discounted rewards and provide respective bounds on the expected discounted regret for a satisficing variant of Thompson sampling.

## 2 The Realizable Case

We start with the *realizable case* when $\mu_* > S$. The main goal of this section is to show that suitable algorithms will have just constant $S$-regret in this case. Note that this does not hold for standard algorithms like UCB1. Lower bounds show that these algorithms will choose a suboptimal arm $i$ for $\Omega\left(\frac{\log T}{(\mu^* - \mu_i)^2}\right)$ times. This of course also holds for any arm below the satisfaction level $S$ giving a contribution to the overall $S$-regret of $\Omega\left(\frac{(S - \mu_i)\log T}{(\mu^* - \mu_i)^2}\right)$.

### 2.1 Simple Algorithm

For the realizable case we propose the simple algorithm SIMPLE-SAT shown as Algorithm 1. It exploits, i.e., plays the empirical best arm so far if its empirical mean reward is $\geqslant S$ and explores uniformly at random otherwise. In the following, the empirical reward for arm $i$ at step $t$ (i.e., *before* choosing the arm $A_t$) is denoted by $\hat{\mu}_i(t)$.

---

**Algorithm 1**: SIMPLE-SAT (Simple Algorithm for Satisficing in the Realizable Case)

---

**Require:** $K$, $S$
1: Play each arm once, i.e., for time steps $t = 1, \ldots, K$ play arm $A_t = t$.
2: **for** time steps $t = K + 1, \ldots$ **do**
3:     **if** $\exists i \, \hat{\mu}_i(t) \geqslant S$ **then**
4:         Play $A_t \leftarrow \arg\max_{i \in [\![1,K]\!]} \hat{\mu}_i(t)$.
5:     **else**
6:         Choose $A_t$ uniformly at random from $[\![1, K]\!]$.
7:     **end if**
8: **end for**

---

While SIMPLE-SAT is quite a natural algorithm, it is in fact based on an idea due to Bubeck et al. (2013) who considered ordinary regret bounds for the MAB setting under the assumption that the learner knows the value of $\mu_*$ as well as (a bound on) the gap $\Delta$ between the optimal and the best suboptimal arm. This knowledge is actually used to compute a reference value $\mu$ that separates the optimal from suboptimal arms, that is, $\mu_* > \mu > \mu_i$ for all suboptimal arms $i$. Bubeck et al. (2013) demonstrate that having such a reference value allows to derive constant regret bounds that are independent of the number of steps $T$. In our setting the satisfaction level $S$ constitutes a similar reference value, which in the realizable case separates the good arms from the bad ones. As will be shown in the next section, accordingly we can derive constant bounds on the $S$-regret.

### 2.2 Regret Bound

Analogously to the ordinary MAB setting where the gaps $\Delta_i := \mu_* - \mu_i$ to the optimal arm appear in bounds on the (classic) regret, the gaps $\Delta_i^S = S - \mu_i$ for non-satisficing arms as well as $|\Delta_*^S| = \mu_* - S$ are important parameters describing the difficulty of the problem in the satisficing setting. Indeed, one can show the following bound on the $S$-regret.

**Theorem 1.** *If $S < \mu_*$ then SIMPLE-SAT satisfies for all $T \geqslant 1$,*

$$R_T^S \leqslant \sum_{i:\Delta_i^S > 0} \left( \Delta_i^S + \frac{2}{\Delta_i^S} + \frac{2\Delta_i^S}{|\Delta_*^S|^2} \right). \tag{1}$$

As the algorithm, the analysis is also based on the ideas of Bubeck et al. (2013). The proof of Theorem 1 is given in Appendix A. In the following, we would like to discuss the result in detail.

**Comparison to other bounds**   For the satisficing setting Reverdy et al. (2017) have proposed a variant of the UCL algorithm (Reverdy et al., 2014) that picks an arbitrary arm with UCL-index above $S$ (instead of an arm with maximal index). For this algorithm they show bounds on their expected satisficing regret that are logarithmic in $T$. As already mentioned, our notion of $S$-regret is an upper bound on the expected satisficing regret of Reverdy et al. (2017), so that our bound, which is independent of $T$, can be considered an improvement. We note however that the setting of Reverdy et al. (2017) is still Bayesian so that the bounds are not directly comparable. Only when considering the slightly different notion of expected satisficing regret in the earlier (Reverdy et al., 2016) the settings as well as the notions of regret coincide.

Reverdy et al. (2017) also claim a lower bound that is logarithmic in the horizon (not mentioned in the corrections of Reverdy et al., 2021), which contradicts Theorem 1. This bound is obtained by application of a lower bound for the *multiple play* setting (Anantharam et al., 1987), where at each step $m$ arms are chosen by the learner, who hence has to identify the $m$ best arms. The given proof chooses $m$ to be all arms above the given satisfaction level $S$. However, the lower bound is not directly applicable to the satisficing setting: Here not *all* arms above the satisfaction level have to be found, a single one is sufficient.

**Lower bounds**   Bubeck et al. (2013) have shown that in the setting when the optimal average reward $\mu_*$ and the gap $\Delta$ are known, the regret for any suboptimal arm with distance $\Delta$ to $\mu_*$ has to be of order $\frac{1}{\Delta}$. This result can be transferred to the satisficing setting to obtain the following lower bound.

**Proposition 1.** *For any algorithm there is a problem setting in which it suffers $S$-regret at least*

$$R_T^S \geqslant \sum_{i:\Delta_i^S > 0} \frac{c}{\Delta_i^S}$$

*for a global constant $c > 0$.*

This means that the second term in the bound of Theorem 1 is unavoidable. Any algorithm has to suffer that much regret in the worst case in order to identify an arm below $S$ to be not satisficing. Unlike that, it is not clear whether the third term involving the quantity $\Delta_*^S$ is necessary. Intuitively, the problem becomes harder when the optimal arm is closer to $S$. However, the bound of Theorem 1 is of the same order as the lower bound in Proposition 1 only when $|\Delta_*^S| \geqslant \Delta_i^S$ for some $i$.

**Extensions**   The methods of Bubeck et al. (2013) only work when the reference value $S$ is below $\mu_*$. That is, they do not give constant regret when $S = \mu_*$. However, in the meantime it has been shown that in the original setting of Bubeck et al. (2013) knowledge of $\mu_*$ is sufficient for obtaining constant bounds on the ordinary regret, that is, neither the gap $\Delta$ nor a reference value strictly below $\mu_*$ are necessary, cf. Appendix C of (Garivier et al., 2019). Accordingly, an adaptation of the algorithm proposed by Garivier et al. (2019) will obtain constant $S$-regret also for the case $S = \mu_*$.

Bubeck et al. (2013) also provide another algorithm with a more refined approach for exploration, using a potential function instead of a uniform probability distribution over the arms. A respective adaptation of algorithm and analysis to the satisficing setting can be done in quite a straightforward way and is given in Appendix C.

## 3   The General Case

Now let us consider the general case where it is not guaranteed that the chosen satisfaction level $S$ is realizable, that is, it may happen that $S > \mu_*$. Then unlike in the realizable case the satisfaction level $S$ does not give the learner any useful information so that we cannot hope to perform better than in the ordinary MAB setting. Obviously the $S$-regret will be linear, but we can still aim at getting bounds on the (classic) regret. On the other hand, if there is at least one arm above the satisfaction level $S$, we would like to re-establish constant bounds on the $S$-regret as in the realizable case.

### 3.1 Algorithm Sat-UCB

The basic idea for designing an algorithm that has constant $S$-regret in the realizable case and classic logarithmic regret bounds otherwise is as follows. On the one hand, it is natural to let such an algorithm exploit whenever there is an arm that is empirical above $S$. Otherwise it seems intuitive to let the algorithm just employ an ordinary bandit algorithm such as UCB1 for which logarithmic regret bounds are known to hold. While we present such a two-phase algorithm (called Sat-UCB$^+$) in Section 4, unfortunately we were not able to show constant $S$-regret for it. In order to achieve that we had to add an additional exploration phase that similar to Simple-SAT explores uniformly at random as long as there is hope that an arm might be above the satisfaction level $S$.

The arising Sat-UCB scheme is shown as Algorithm 2. Sat-UCB exploits when there is an arm with empirical mean above $S$ (cf. line 4 of the algorithm) and explores otherwise. The exploration takes into account a UCB value similar to the classical index suggested for the UCB1 algorithm of (Auer et al., 2002), that is,

$$\text{UCB}_i(t) := \hat{\mu}_i(t) + \beta_i(t), \text{ where } \beta_i(t) = \sqrt{\tfrac{2\log(f(t))}{n_i(t-1)}}. \tag{2}$$

Here and in the following, we choose $f(t) = 1 + t\log^2(t)$ and let $n_i(t)$ denote the number of times arm $i$ has been played after step $t$. If there is at least one arm with UCB-value above $S$ then Sat-UCB chooses such an arm uniformly at random, which makes sure that all promising arms are explored sufficiently to decide whether they are satisficing. Otherwise, if all arms have UCB-value below $S$, the algorithm chooses an arm according to UCB1, that is, an arm $i$ maximizing $\text{UCB}_i$. This guarantees that the algorithm performs similar to UCB1 when there is no satisficing arm.

---

**Algorithm 2**: Sat-UCB Scheme for Satisficing in the General Case

**Require:** $K, S$
 1: Play each arm once, i.e., for time steps $t = 1, \ldots, K$ play arm $A_t = t$.
 2: **for** time steps $t = K + 1, \ldots$ **do**
 3:     **if** $\exists i \, \hat{\mu}_i(t) \geqslant S$ **then**
 4:         Choose an arbitrary $A_t$ from $\{i \,|\, \hat{\mu}_i(t) \geqslant S\}$.
 5:     **else if** $\exists i \, \text{UCB}_i(t) \geqslant S$ **then**
 6:         Choose $A_t$ uniformly at random from $\{i \,|\, \text{UCB}_i(t) \geqslant S\}$.
 7:     **else**
 8:         Play arm $A_t \in \underset{i \in [\![1,K]\!]}{\arg\max} \, \text{UCB}_i(t)$ .
 9:     **end if**
10: **end for**

---

In line 4 of Sat-UCB different concrete instantiations for this exploitation step are possible. In Section 4 we will consider different sub-algorithms for choosing an arm from $\{i \,|\, \hat{\mu}_i(t) \geqslant S\}$. The regret bounds we give below for Sat-UCB are independent of the selected exploitation sub-algorithm.

### 3.2 Regret Bounds

In the realizable case, Sat-UCB achieves constant regret similar to Theorem 1 for Simple-SAT.

**Theorem 2.** *If $\mu_* > S$ then Sat-UCB satisfies for all $T \geqslant 1$,*

$$R_T^S \leqslant \sum_{i:\Delta_i^S > 0} \left( \Delta_i^S + \frac{2}{\Delta_i^S} + \frac{7\Delta_i^S}{|\Delta_*^S|^2} \right).$$

On the other hand, when the satisfaction level cannot be reached, we obtain a logarithmic bound on the regret just as for UCB1 (Auer et al., 2002).

**Theorem 3.** *If $\mu_* \leqslant S$ then SAT-UCB satisfies for all $T \geqslant 1$,*

$$R_T \leqslant \sum_{i:\Delta_i>0} \inf_{\varepsilon\in(0,\Delta_i)} \Delta_i \left(1 + \frac{5}{\varepsilon^2} + \frac{2(\log f(T) + \sqrt{\pi \log f(T)} + 1)}{(\Delta_i - \varepsilon)^2}\right). \tag{3}$$

*Furthermore,*

$$\limsup_{T\to\infty} \frac{R_T}{\log(T)} \leqslant \sum_{i:\Delta_i>0} \frac{2}{\Delta_i}. \tag{4}$$

*Thus, for a global constant $C > 0$ it holds that*

$$R_T \leqslant C \sum_{i:\Delta_i>0} \left(\Delta_i + \frac{\log(T)}{\Delta_i}\right).$$

**Discussion** Theorems 2 and 3 show that SAT-UCB is able to achieve the best of both worlds, that is, constant regret in the realizable case and standard logarithmic regret bounds otherwise. The only comparable result in the literature we are aware of are the bounds for lenient regret given by Merlis & Mannor (2021). As already mentioned, the *lenient regret* with the hinge loss as $\varepsilon$-gap function corresponds to $S$-regret when choosing $S := \mu^* - \varepsilon$. Merlis & Mannor (2021) give asymptotic logarithmic bounds on the lenient regret and show that for the special case when $\mu_* > 1 - \varepsilon$ the lenient regret is constant, which resembles the results we have for $S$-regret. However, we note that the results are not equivalent: The lenient regret is with respect to the value $\mu^* - \varepsilon$ and constant regret is only obtained when $\mu^*$ itself is $\varepsilon$-close to the theoretical maximum reward 1. Unlike that, considering an absolute satisfaction level $S$ we can guarantee constant $S$-regret whenever $\mu^* > S$. This holds in particular when $S$ is chosen to be $\mu^* - \varepsilon$ and without further assumptions on $\mu^*$.

### 3.3 Proofs

### 3.3.1 Proof of Theorem 2

We write the $S$-regret as

$$R_T^S = \sum_{i:\Delta_i^S>0}^k \mathbb{E}(n_i(T)) \Delta_i^S$$

and proceed bounding $\mathbb{E}(n_i(T)) = \sum_{t=1}^T \mathbb{P}(A_t = i)$ for all non-satisfying arms $i$. Thus let $i$ be the index of a non-satisfying arm. Let $Z_t := \{\forall j \in [\![1, K]\!], \hat{\mu}_j(t) < S\}$ be the event that all arms have empirical values below the satisfaction level. Now let us consider all possible cases in which arm $i$ is played by the algorithm: After step $K$ this happens either when it has empirical mean above $S$ (line 4 of the algorithm) or if its UCB-value is above $S$ (lines 6 and 8). In the latter case there cannot be any arm with empirical mean above $S$ (event $Z_t$). Accordingly, we can decompose the event $\{A_t = i\}$ as

$$\begin{aligned}
\{A_t = i\} \subset &\{t = i\} \cup \{A_t = i \text{ and } \hat{\mu}_i(t) \geqslant S \text{ and } t > K\} \\
&\cup \{A_t = i \text{ and } \text{UCB}_i(t) \geqslant S \text{ and } \text{UCB}_*(t) \geqslant S \text{ and } t > K \text{ and } Z_t\} \\
&\cup \{A_t = i \text{ and } \text{UCB}_*(t) < S \text{ and } t > K \text{ and } Z_t\}.
\end{aligned} \tag{5}$$

For the first event we have

$$\sum_{t=1}^T \mathbb{P}(t = i) \leqslant 1. \tag{6}$$

For the second event, we obtain analogously to eq. 14 in the proof of Theorem 1 in Appendix A, now writing $\hat{\mu}_{i,n}$ for the empirical estimate of $\mu_i$ computed from $n$ samples,

$$
\begin{aligned}
\sum_{t=1}^{T} \mathbb{P}\big(A_t = i, \hat{\mu}_i(t) \geqslant S, t > K\big) &\leqslant \sum_{t=1}^{T} \mathbb{P}\big(A_t = i, \hat{\mu}_i(t) \geqslant \mu_i + \Delta_i^S\big) \\
&\leqslant \sum_{n=1}^{T} \mathbb{P}(\hat{\mu}_{i,n} \geqslant \mu_i + \Delta_i^S) \\
&\leqslant \frac{2}{(\Delta_i^S)^2}.
\end{aligned}
\tag{7}
$$

For the probability of the third event we have, again similar to the derivation of eq. 14, now using $*$ to refer to an arbitrary optimal arm,

$$
\begin{aligned}
\sum_{t=1}^{T} \mathbb{P}\big(A_t = i, \mathrm{UCB}_i(t) &\geqslant S, \mathrm{UCB}_*(t) \geqslant S, t > K, Z_t\big) \\
&= \sum_{t=1}^{T} \mathbb{P}\big(A_t = *, \mathrm{UCB}_i(t) \geqslant S, \mathrm{UCB}_*(t) \geqslant S, t > K, Z_t\big) \\
&\leqslant \sum_{t=1}^{T} \mathbb{P}\big(A_t = *, Z_t\big) \leqslant \sum_{t=1}^{T} \mathbb{P}\big(A_t = *, \hat{\mu}_*(t) \leqslant S\big) = \mathbb{E}\Big(\sum_{t=1}^{T} \mathbb{1}\big\{A_t = *, \hat{\mu}_*(t) \leqslant S\big\}\Big) \\
&\leqslant \mathbb{E}\Big(\sum_{n=1}^{T} \mathbb{1}\{\hat{\mu}_{*,n} \leqslant S\}\Big) = \sum_{n=1}^{T} \mathbb{P}(\hat{\mu}_{*,n} \leqslant S) = \sum_{n=1}^{T} \mathbb{P}\big(\hat{\mu}_{*,n} \leqslant \mu_* - |\Delta_*^S|\big) \\
&\leqslant \sum_{n=1}^{T} \exp\big(-\tfrac{n|\Delta_*^S|^2}{2}\big) \\
&\leqslant \frac{2}{|\Delta_*^S|^2}.
\end{aligned}
\tag{8}
$$

Finally, the probability of the last event of eq. 5 is upper bounded by

$$
\begin{aligned}
\sum_{t=1}^{T} \mathbb{P}\big(\mathrm{UCB}_*(t) < S\big) &= \sum_{t=1}^{T} \mathbb{P}\big(\hat{\mu}_*(t) < \mu_* - (|\Delta_*^S| + \beta_*(t))\big) \\
&\leqslant \sum_{t=1}^{T} \sum_{n=1}^{t} \mathbb{P}\left(\hat{\mu}_{*,n} < \mu_* - \Big(|\Delta_*^S| + \sqrt{\tfrac{2\log(f(t))}{n}}\Big)\right) \\
&\leqslant \sum_{t=1}^{T} \sum_{n=1}^{t} \frac{1}{f(t)} \exp\big(-\tfrac{n|\Delta_*^S|^2}{2}\big) \leqslant \frac{2}{|\Delta_*^S|^2} \sum_{t=1}^{T} \frac{1}{f(t)} \\
&\leqslant \frac{5}{|\Delta_*^S|^2},
\end{aligned}
\tag{9}
$$

where the last inequality is obtained by observing that $\sum_{t=1}^{T} \frac{1}{f(t)} \leqslant 1 + \sum_{t=2}^{T} \frac{1}{t\log^2(t)}$ and then bounding the sum with an integral.

Putting everything together, we obtain from equations eqs. 5–9 the claimed result

$$
R_T^S = \sum_{i:\Delta_i^S > 0} \Delta_i^S \, \mathbb{E}(n_i(T)) \leqslant \sum_{i:\Delta_i^S > 0} \Big(\Delta_i^S + \frac{2}{\Delta_i^S} + \frac{7\Delta_i^S}{|\Delta_*^S|^2}\Big).
$$

$\square$

### 3.3.2 Proof of Theorem 3

The proof of Theorem 3 can be reduced to the derivation of the regret bounds for UCB1 as given in Theorem 8.1 of Lattimore & Szepesvári (2020). We start with the standard regret decomposition

$$R_T = \sum_{i:\Delta_i>0} \mathbb{E}\left(n_i(T)\right)\Delta_i.$$

In the following, we bound for each suboptimal arm $i$ the number of times $n_i(T)$ it is played. By definition of the algorithm, arm $i$ is chosen after step $K$ only if either

$$\hat{\mu}_i(t) + \beta_i(t) \geqslant \hat{\mu}_*(t) + \beta_*(t) \quad \text{or} \quad \hat{\mu}_i(t) + \beta_i(t) \geqslant S.$$

(Note that the case $\hat{\mu}_i(t) \geqslant S$ is subsumed by the second event.) Accordingly, we can decompose the event $A_t = i$ using some arbitrary but fixed $\varepsilon \in (0, \Delta_i)$ as

$$
\begin{aligned}
\{A_t = i\} \subseteq &\big\{A_t = i \text{ and } \hat{\mu}_*(t) + \beta_*(t) \leqslant \mu_* - \varepsilon\big\} \\
&\cup \big\{A_t = i \text{ and } \hat{\mu}_*(t) + \beta_*(t) \geqslant \mu_* - \varepsilon\big\} \\
\subseteq &\big\{\hat{\mu}_*(t) + \beta_*(t) \leqslant \mu_* - \varepsilon\big\} \\
&\cup \big\{A_t = i \text{ and } \hat{\mu}_i(t) + \beta_i(t) \geqslant \hat{\mu}_*(t) + \beta_*(t) \geqslant \mu_* - \varepsilon\big\} \\
&\cup \big\{A_t = i \text{ and } \hat{\mu}_*(t) + \beta_*(t) \geqslant \mu_* - \varepsilon \text{ and } \hat{\mu}_i(t) + \beta_i(t) \geqslant S\big\} \\
\subseteq &\big\{\hat{\mu}_*(t) + \beta_*(t) \leqslant \mu_* - \varepsilon\big\} \\
&\cup \big\{A_t = i \text{ and } \hat{\mu}_i(t) + \beta_i(t) \geqslant \mu_* - \varepsilon\big\},
\end{aligned}
$$

where the last inclusion is due to the assumption that $S \geqslant \mu_*$. It follows that

$$n_i(T) \leqslant \sum_{t=1}^{T} \mathbb{1}\big\{\hat{\mu}_*(t) + \beta_*(t) \leqslant \mu_* - \varepsilon\big\} + \sum_{t=1}^{T} \mathbb{1}\big\{A_t = i \text{ and } \hat{\mu}_i(t) + \beta_i(t) \geqslant \mu_* - \varepsilon\big\}.$$

The obtained decomposition is the same as the one in the proof of Theorem 8.1 from (Lattimore & Szepesvári, 2020) and the very same arguments can be used to finish the proof of eq. 3. The second part of the theorem, that is eq. 4, follows by choosing $\varepsilon = \log^{-1/4}(T)$ and taking the limit as $T$ tends to infinity. $\qquad\square$

## 4 Experiments

We compared Sat-UCB to other bandit algorithms in order to show that the latter keep accumulating $S$-regret, while Sat-UCB sticks to a satisficing arm after finite time, thus confirming the results of Theorem 2. We also investigated the behavior of Sat-UCB in the not realizable case with different values for the chosen satisfaction level $S$ and did experiments with a slightly modified version Sat-UCB$^+$ introduced below in Section 4.2.

### 4.1 Exploitation in Sat-UCB

As already mentioned, we investigated different sub-algorithms for exploitation in line 4 of Sat-UCB. Obvious choices for this exploitation step are e.g. selecting the arm with maximal empirical mean reward or using UCB1 to choose among the arms with empirical mean reward above $S$.

However, the following more refined approach for exploitation empirically worked best. In addition to the UCB value for each arm we define an analogous lower confidence bound value

$$\text{LCB}_i(t) := \hat{\mu}_i(t) - \beta_i(t). \tag{10}$$

Then for any arm $i$ with empirical mean above $S$ we consider the confidence interval $[LCB_i, UCB_i]$ and then choose the arm for which the largest share of this confidence interval is above the satisfaction level $S$. That

is, in line 4 SAT-UCB chooses an arm from

$$\underset{i\in[\![1,K]\!]}{\operatorname{argmax}} \left\{ \frac{\text{UCB}_i(t) - \max\{S,\, \text{LCB}_i(t)\}}{\beta_i(t)} \right\}. \tag{11}$$

The intuition behind this choice is that an arm whose confidence interval ist mostly above $S$ will most likely have actual mean above $S$. In the following experiments, *SAT-UCB* always refers to SAT-UCB employing this *confidence fraction* index as exploitation sub-algorithm.

## 4.2 Sat-UCB$^+$: Modified Exploration in Sat-UCB

Concerning exploration we also considered a simplified version of SAT-UCB which does not use random exploration and instead always plays UCB1 when there is no empirically satisfying arm. For the sake of completeness, this modification SAT-UCB$^+$ is shown as Algorithm 3.

---

**Algorithm 3**: SAT-UCB$^+$ (Experimental Simplification of SAT-UCB)

---

**Require:** $K, S$

1: Play each arm once, i.e., for time steps $t = 1, \ldots, K$ play arm $A_t = t$.
2: **for** time steps $t = K + 1, \ldots$ **do**
3:      **if** $\exists i \, \hat{\mu}_i(t) \geqslant S$ **then**
4:          Choose $A_t$ from $\underset{i\in[\![1,K]\!]}{\operatorname{argmax}} \left\{ \frac{\text{UCB}_i(t) - \max\{S,\, \text{LCB}_i(t)\}}{\beta_i(t)} \right\}$.
5:      **else**
6:          Play arm $A_t \in \underset{i\in[\![1,K]\!]}{\operatorname{argmax}} \text{UCB}_i(t)$ .
7:      **end if**
8: **end for**

---

While we were not able to provide a constant bound on the $S$-regret as for the original SAT-UCB algorithm, in the experiments SAT-UCB$^+$ performed better than SAT-UCB.

## 4.3 Experimental Setup

### 4.3.1 Settings

To illustrate the influence of the structure of the underlying bandit problem we performed experiments in the following two settings each with 20 arms and normally distributed rewards[1] with standard deviation 1:

In *Setting 1* the mean reward of each arm $i = 1, 2, \ldots, 20$ is set to $\frac{i-1}{20}$. For the satisfaction level we chose 0.8 in the realizable case, resulting in four satisfying arms. For experiments in the not realizable case $S = 1$ was chosen.

In *Setting 2* the mean reward of each arm $i$ is set to $\sqrt{\frac{i}{20}}$. For the satisfaction level we chose $S = 0.92$ so that there are three satisfying arms in the realizable case. This setting is more difficult than Setting 1, as arms are closer to $S$ as well as to each other. The not realizable satisfaction level was set to 1.1.

Further experiments for a complementary very simple setting, in which only has to be decided between an arm giving reward 1 and arms with reward 0, are reported in Appendix B.

### 4.3.2 Algorithms

We compared different instantiations of SAT-UCB. Beside the main variant that uses the confidence fraction index in eq. 11 for exploitation in line 4 of SAT-UCB, we also consider using UCB1 or the maximal empirical

---

[1]We repeated all experiments also with Bernoulli rewards, which gave similar results, which are consequently not reported here.

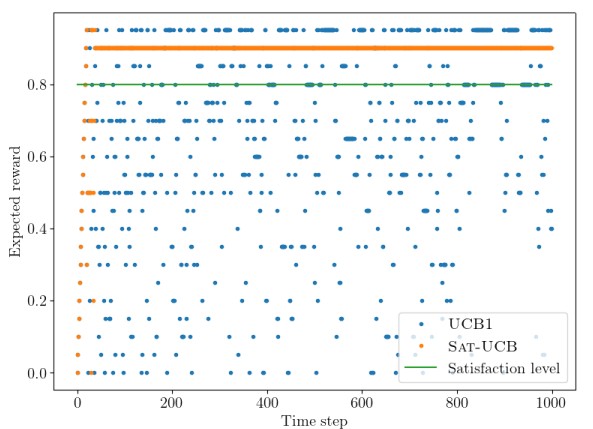

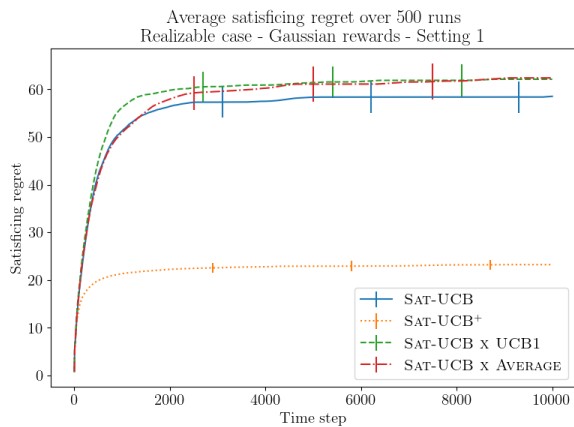

(a) Arm pulls for UCB1 and SAT-UCB in exemplary run.

(b) Comparing the $S$-regret for variants of SAT-UCB.

Figure 1: Experiments in the realizable case of Setting 1.

mean for choosing an arm, respectively. We also performed experiments with the modified SAT-UCB$^+$ algorithm.

For comparison we considered UCB1 (choosing the same confidence intervals as for SAT-UCB) as well as other algorithms with focused exploitation, that is, UCB$_\alpha$ (Degenne et al., 2019), the Satisfaction in Mean Reward UCL algorithm (Reverdy et al., 2017), and the (deterministic) UCL algorithm (Reverdy et al., 2014) it is based on.

For UCB$_\alpha$ that aims to combine regret minimization and best arm identification we chose confidence $\delta = 0.001$ and $\alpha = 1$ to focus on exploitation. We also performed a few experiments with $\alpha > 1$ that confirmed that in this case the algorithm explores more and with increasing $\alpha$ accumulates more $S$-regret. The original UCB$_\alpha$ is given for just two arms and stops when the optimal arm has been identifed. In our case with an arbitrary number of arms we eliminate arms that are identified as suboptimal with respect to same criterion as suggested by Degenne et al. (2019).

For Satisfaction in Mean Reward UCL we considered various ways of how to choose arms from the *eligible set* (cf. eq. 28 of Reverdy et al., 2017), as this step is not specified in the original paper. Experimentally it worked best to select at any step $t$ *each* arm from the eligible set once but increase the time step counter just by 1, independent of how many arms have been chosen at $t$. For the parameter $a$ we chose $a = 1$, which was suggested in the original paper and worked best experimentally, although Reverdy et al. (2021) state that for the theoretical results to hold one should have $a > \frac{4}{3}$.

### 4.4 Results

#### 4.4.1 Realizable Case

We started with comparing SAT-UCB to UCB1 in Setting 1. Figure 1a depicts a showcase run illustrating that SAT-UCB soon focuses on a satisficing arm, while UCB1 keeps exploring. Accordingly, UCB1 suffers growing $S$-regret due to ongoing exploration of arms below the satisfaction level, cf. Figure 2 below.

Figure 1b shows a comparison of the $S$-regret of the different versions of SAT-UCB as well as the experimental modification SAT-UCB$^+$ in Setting 1. Here and in the following plots, we show the averaged values for ($S$-)regret over 100 runs and the error bars (which are sometimes barely visible) indicate an estimate of the standard error (i.e., the sample standard deviation of the cumulated regret over the square root of the number of respective runs). We see that the experimental modification SAT-UCB$^+$ not only achieves the smallest $S$-regret, it also displays much smaller standard error than the SAT-UCB variants. Experiments for Setting 2

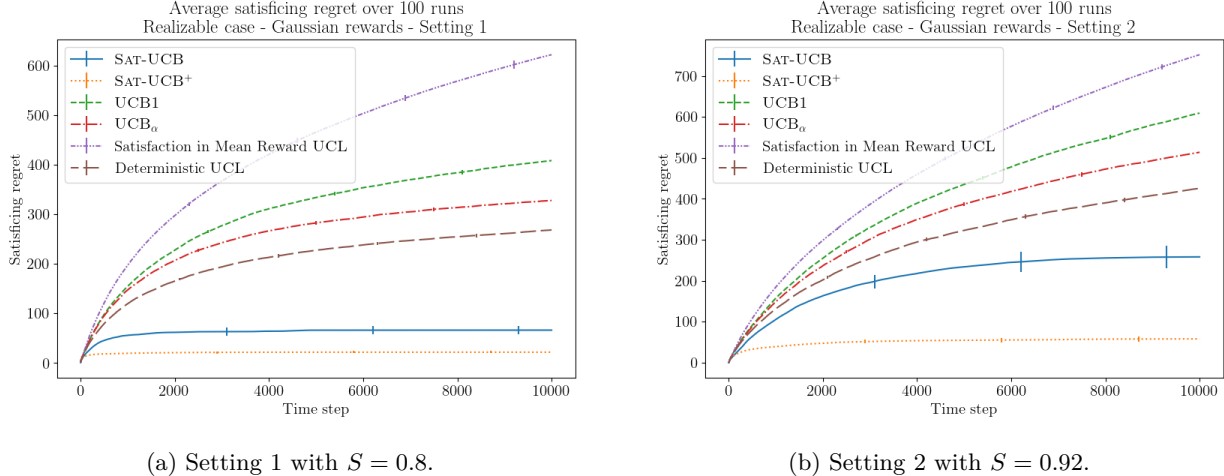

(a) Setting 1 with $S = 0.8$.             (b) Setting 2 with $S = 0.92$.

Figure 2: Comparison of $S$-regret of different algorithms in the realizable case.

give similar results (cf. Fig. 2b below). We note that for SAT-UCB$^+$ the confidence fraction index of eq. 11 also gives better results than using UCB1 or choosing the empirically best arm instead.

Figure 2 shows a comparison of the $S$-regret of SAT-UCB and SAT-UCB$^+$ to other algorithms. Although $S$-regret is smaller than classic regret, all algorithms except our two algorithms suffer growing regret due to ongoing exploration of arms below the satisfaction level. As expected, SAT-UCB gives constant regret, while surprisingly Deterministic UCL is superior to its Satisfaction in Mean Reward counterpart. Figure 2b illustrates that the regret for SAT-UCB is larger in Setting 2 in which the gaps of the relevant arms to the satisfaction level are smaller. In both cases SAT-UCB$^+$ performs best however.

### 4.4.2 Not Realizable Case

In the not realizable case SAT-UCB usually performed a bit worse than UCB1, cf. Figure 3a. (The respective plot for Setting 2 can be found in Figure 4a of Appendix B.) In particular, in the beginning SAT-UCB does more (random) exploration and catches up only for larger horizon. The experimental modification SAT-UCB$^+$ shows the opposite behavior performing much better for small horizon before performance coincides with UCB1 and SAT-UCB after a higher number of steps.

Interestingly, while SAT-UCB is quite insensitive to the choice of $S$, in the not realizable case SAT-UCB$^+$ works better the closer $S$ is chosen to $\mu_*$. This is illustrated in Figure 3b. When $S$ is chosen close to $\mu_* = 0.95$ the regret becomes nearly constant. This behavior of SAT-UCB$^+$ can be explained as follows: When $S$ is close to $\mu^*$ this increases exploitation in case there are arms with empirical mean above $S$. With $S$ being close to $\mu^*$ it is also more likely that such arms exist. On the other hand, if there are no such arms the increased exploitation when using UCB1 (instead of random exploration as in SAT-UCB) leads to improved performance of SAT-UCB$^+$.

## 5 Conclusion

Our results for the multi-armed bandit case are just a first step in an ongoing project on satisficing in reinforcement learning. While some ideas may be used also in the general standard Markov decision process setting, it seems already not quite simple to obtain reasonable constant regret bounds in the realizable case. While it might be possible to consider each policy as an arm in an MAB setting, the resulting bounds would be linear in the number of policies and hence exponential in the number of states.

Another interesting direction for future research is satisficing with adaptive satisfaction level. While overall such an approach obviously will not be possible to obtain constant regret in any case, it is an interesting

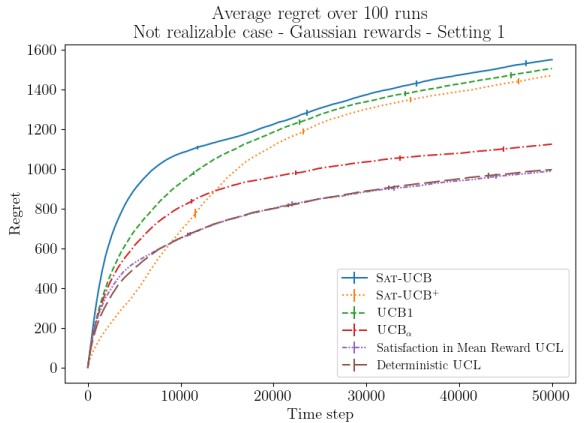
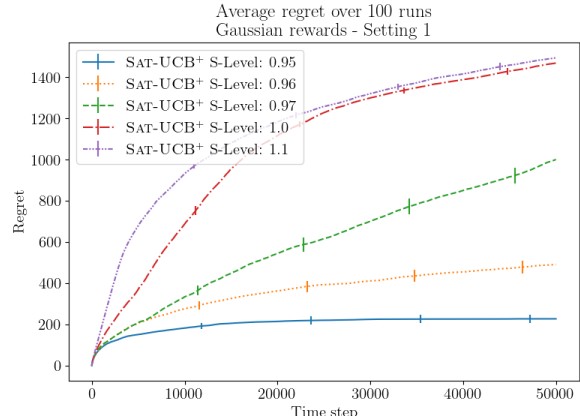

(a) Comparison of classic regret of different algorithms in the not realizable case of Setting 1 with $S = 1$.

(b) Comparison of different choices of S for SAT-UCB$^+$ in Setting 1.

Figure 3: Classic regret in the not realizable case.

question what could be gained by trying to adapt the satisfaction level towards the optimal mean reward. This is in particular interesting in view of the results for SAT-UCB$^+$, although currently provable regret bounds for this version of the algorithm are still missing.

A lesson to take from the MAB setting is that the savings from considering a satisficing instead of an optimizing objective –at least with respect to regret– is not that there are arms that need no exploration at all. Rather in the worst case (as always considered by notions of regret) one still has to explore all arms, however the amount of necessary exploration is now constant and independent of the horizon.

### Acknowledgments

The authors would like to thank the anonymous reviewers for their valuable comments that helped to improve the paper. In particular, we are grateful for one NeurIPS reviewer pointing out an error in the proof of Theorem 2 for a slightly different variant of SAT-UCB (basically, SAT-UCB$^+$) in an earlier version of this paper. This work was supported by the Austrian Science Fund (FWF): TAI 590-N.

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

## A Proof of Theorem 1

For the proof we shall need the following result that follows by our assumption of sub-Gaussianity and a Chernoff bound.

**Lemma 1.** *Let $\hat{\mu}_{i,n}$ be an empirical estimate for $\mu_i$ computed from $n$ samples. Then for all $\varepsilon > 0$ and each $i \in [\![1, K]\!]$,*

$$\mathbb{P}(\hat{\mu}_{i,n} \geqslant \mu_i + \varepsilon) \leqslant \exp(-\tfrac{n\varepsilon^2}{2}),$$
$$\mathbb{P}(\hat{\mu}_{i,n} \leqslant \mu_i - \varepsilon) \leqslant \exp(-\tfrac{n\varepsilon^2}{2}).$$

*Proof of Theorem 1.* Let $i$ be the index of a non-satisficing arm. In the following we decompose the event that arm $i$ is chosen at some step $t$. To do that we introduce the event $Z_t := \{\forall j \in [\![1, K]\!], \hat{\mu}_j(t) < S\}$ that all arms have empirical estimates below $S$, when the algorithm chooses an arm randomly according to line 6 of the algorithm. Then we have

$$\{A_t = i\} \subset \{t = i\} \cup \{A_t = i, Z_t^c\} \cup \{A_t = i, Z_t\}. \tag{12}$$

For the first two events we have

$$\sum_{t=1}^{T} \mathbb{P}(t = i) \leqslant 1 \tag{13}$$

and using Lemma 1

$$\sum_{t=1}^{T} \mathbb{P}(A_t = i, Z_t^c) \leqslant \sum_{t=1}^{T} \mathbb{P}(A_t = i, \hat{\mu}_i(t) \geqslant S) = \sum_{t=1}^{T} \mathbb{P}(A_t = i, \hat{\mu}_i(t) \geqslant \mu_i + \Delta_i^S)$$

$$\leqslant \sum_{n=1}^{T} \mathbb{P}(\hat{\mu}_{i,n} \geqslant \mu_i + \Delta_i^S)$$

$$\leqslant \sum_{n=1}^{T} \exp\left(-\tfrac{n(\Delta_i^S)^2}{2}\right)$$

$$\leqslant \frac{e^{-\frac{(\Delta_i^S)^2}{2}}}{1 - e^{-\frac{(\Delta_i^S)^2}{2}}}$$

$$\leqslant \frac{2}{(\Delta_i^S)^2}. \tag{14}$$

Rewriting the probability of the third event in eq. 12, using $*$ to refer to an arbitrary optimal arm, we obtain

$$\mathbb{P}(A_t = i, Z_t) = \mathbb{P}(A_t = i | Z_t) \, \mathbb{P}(Z_t) = \tfrac{1}{K} \cdot \mathbb{P}(Z_t)$$
$$= \mathbb{P}(A_t = * | Z_t) \, \mathbb{P}(Z_t) = \mathbb{P}(A_t = *, Z_t).$$

Now summing over the time steps up to $T$ yields

$$\sum_{t=1}^{T} \mathbb{P}(A_t = i, Z_t) = \sum_{t=1}^{T} \mathbb{P}(A_t = *, Z_t)$$

$$\leqslant \sum_{t=1}^{T} \mathbb{P}\big(A_t = *, \hat{\mu}_*(t) \leqslant S\big) \; = \; \mathbb{E}\Big(\sum_{t=1}^{T} \mathbb{1}\big\{A_t = *, \hat{\mu}_*(t) \leqslant S\big\}\Big)$$

$$\leqslant \mathbb{E}\Big(\sum_{n=1}^{T} \mathbb{1}\{\hat{\mu}_{*,n} \leqslant S\}\Big) \; = \; \sum_{n=1}^{T} \mathbb{P}(\hat{\mu}_{*,n} \leqslant S)$$

$$= \sum_{n=1}^{T} \mathbb{P}\big(\hat{\mu}_{*,n} \leqslant \mu_* - |\Delta_*^S|\big)$$

$$\leqslant \sum_{n=1}^{T} \exp\big(-\tfrac{n|\Delta_*^S|^2}{2}\big)$$

$$\leqslant \frac{2}{|\Delta_*^S|^2}. \tag{15}$$

Finally writing

$$n_i(T) = \sum_{t=1}^{T} \mathbb{1}\{A_t = i\}$$

for the number of times arm $i$ was pulled up to step $T$, we can combine eqs. 12–15 to obtain

$$R_T^S = \sum_{i:\Delta_i^S>0} \Delta_i^S \, \mathbb{E}(n_i(T))$$

$$= \sum_{i:\Delta_i^S>0} \Delta_i^S \sum_{t=1}^{T} \mathbb{P}(A_t = i)$$

$$\leqslant \sum_{i:\Delta_i^S>0} \Delta_i^S \Big(1 + \frac{2}{(\Delta_i^S)^2} + \frac{2}{|\Delta_*^S|^2}\Big)$$

$$= \sum_{i:\Delta_i^S>0} \Big(\Delta_i^S + \frac{2}{\Delta_i^S} + \frac{2\Delta_i^S}{|\Delta_*^S|^2}\Big). \qquad \square$$

## B  Complementary Experiments

In this section we report additional experiments that we performed in the following very easy *Setting 3*: There are 19 arms with mean reward 0 and one with mean reward 1. Here for any satisfaction level between 0 and 1 the task of satisficing is equivalent to learning the optimal arm. However knowing the satisfaction level gives some additional information that allows to learn faster. For the experiments $S = 0.5$ was chosen in the realizable case and $S = 1.1$ in the not realizable case.

As shown in Fig. 4b, in the realizable case SAT-UCB$^+$ and SAT-UCB work equally well in this very simple setting. Also UCB$_\alpha$ exhibits at least close to constant regret in Setting 3. In the not realizable case the two UCL variants perform best, cf. Figure 5a. As Figure 5b demonstrates, SAT-UCB$^+$ performs better the closer $S$ is chosen to $\mu_*$.

## C  Exploration Based on a Potential Function

Following Bubeck et al. (2013), Algorithm 4 presents a more general approach for exploration, using a potential function $\psi : [0, \infty) \to \mathbb{R}^+$ that is assumed to be differentiable and increasing.

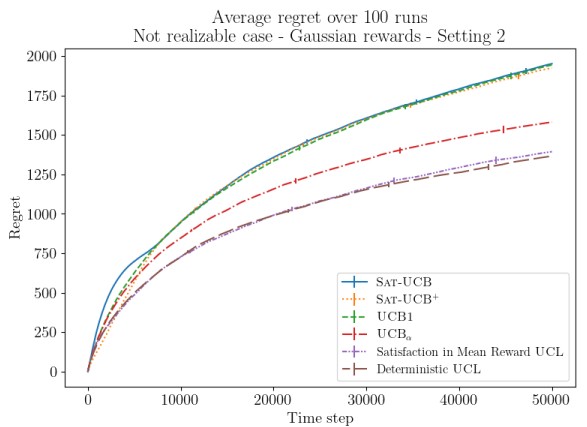
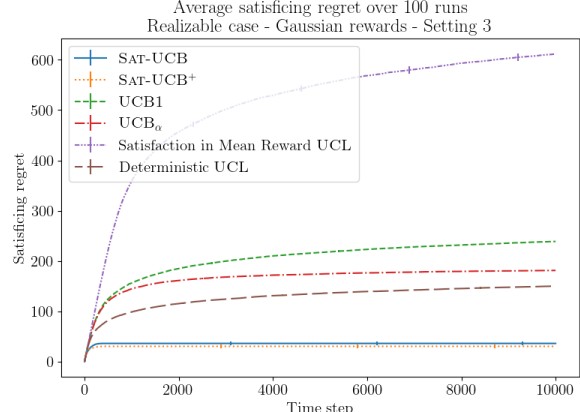

(a) Classic regret in the not realizable Setting 2 with $S = 1.1$.

(b) $S$-regret for the realizable case in Setting 3 with $S$=0.5.

Figure 4

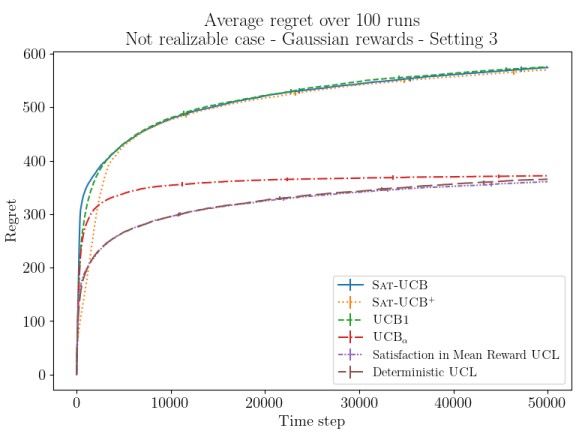
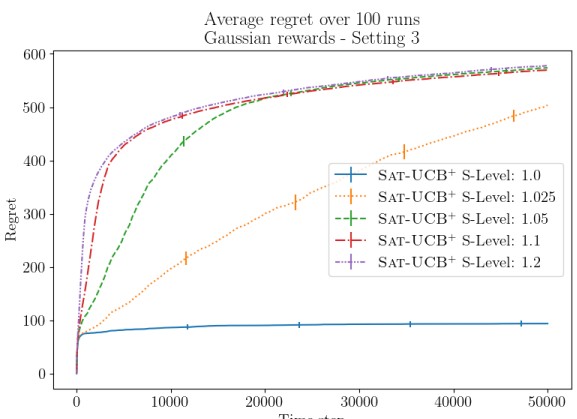

(a) Comparison of algorithms in Setting 3 with $S$=1.1.

(b) Comparison of different choices of $S$ for SAT-UCB$^+$.

Figure 5: Classic regret in the not realizable Setting 3.

---

**Algorithm 4**: POT-SAT (Algorithm for Satisficing in the Realizable Case Using a Potential Function)

---

**Require:** $K$, $S$

1: Play each arm once, i.e., for time steps $t = 1, \ldots, K$ play arm $A_t = t$.
2: **for** time steps $t = K + 1, \ldots$ **do**
3:    **if** $\exists i \, \hat{\mu}_i(t) \geqslant S$ **then**
4:        Play $A_t \leftarrow \arg\max_{i \in [\![1,K]\!]} \hat{\mu}_i(t)$.
5:    **else**
6:        Choose an arm randomly according to the probability distribution defined by

$$p_{i,t} = \frac{1}{\alpha \cdot \psi(|S - \hat{\mu}_i(t)|)},$$

7:        where $\alpha = \sum_{j=1}^{K} \frac{1}{\psi(|S - \hat{\mu}_j(t)|)}$.
8:    **end if**
9: **end for**

---

**Theorem 4.** *Let $\psi : [0, \infty) \to \mathbb{R}^+$ be a differentiable and increasing function. If $\mu_* > S$ then Algorithm 4 satisfies for all $T \geqslant 1$,*

$$R_T^S \leqslant \sum_{i:\Delta_i^S > 0} \left( \Delta_i^S + \frac{8}{\Delta_i^S} + \frac{\Delta_i^S}{\psi(\frac{\Delta_i^S}{2})} \left( \frac{2\psi(0)}{(\Delta_*^S)^2} + \int_0^{+\infty} \frac{\psi'(x)}{e^{\frac{(|\Delta_*^S| + x)^2}{2}} - 1} \, dx \right) \right).$$

*Proof.* As in the proof of Theorem 1 we aim at a bound on $\mathbb{E}(n_i(T)) = \sum_{t=1}^{T} \mathbb{P}(A_t = i)$ for each non-satisficing arm $i$. First, we decompose the event $\{A_t = i\}$ as

$$\{A_t = i\} \subset \{t = i\} \cup \left\{ A_t = i, \hat{\mu}_i(t) > S - \tfrac{\Delta_i^S}{2}, t > K \right\}$$
$$\cup \left\{ A_t = i, \hat{\mu}_i(t) \leqslant S - \tfrac{\Delta_i}{2}, t > K \right\}. \tag{16}$$

For the first two events we have

$$\sum_{t=1}^{T} \mathbb{P}(t = i) \leqslant 1 \tag{17}$$

and

$$\sum_{t=1}^{T} \mathbb{P}\left\{ A_t = i, \hat{\mu}_i(t) > S - \tfrac{\Delta_i^S}{2}, t > K \right\} \leqslant \frac{8}{(\Delta_i^S)^2}.$$

For the probability of the third event in eq. 16 we have

$$\begin{aligned}
\mathbb{P}\left( A_t = i, \hat{\mu}_i(t) \leqslant S - \tfrac{\Delta_i^S}{2}, t > K \right) &\leqslant \mathbb{P}\left( A_t = i, \hat{\mu}_i(t) \leqslant S - \tfrac{\Delta_i^S}{2}, Z_t \right) \\
&= \mathbb{E}\left( p_{i,t} \mathbb{1}\left\{ \hat{\mu}_i(t) \leqslant S - \tfrac{\Delta_i^S}{2}, Z_t \right\} \right) \\
&= \mathbb{E}\left( \frac{p_{i,t}}{p_{*,t}} p_{*,t} \mathbb{1}\left\{ \hat{\mu}_i(t) \leqslant S - \tfrac{\Delta_i^S}{2}, Z_t \right\} \right) \\
&\leqslant \mathbb{E}\left( \frac{\psi(|S - \hat{\mu}_*(t)|)}{\psi(\frac{\Delta_i^S}{2})} p_{*,t} \mathbb{1}\left\{ \hat{\mu}_i(t) \leqslant S - \tfrac{\Delta_i^S}{2}, Z_t \right\} \right) \\
&\leqslant \frac{1}{\psi(\frac{\Delta_i^S}{2})} \mathbb{E}\left( \psi(|S - \hat{\mu}_*(t)|) \, p_{*,t} \, \mathbb{1}\{Z_t\} \right) \\
&\leqslant \frac{1}{\psi(\frac{\Delta_i^S}{2})} \mathbb{E}\left( \psi(|S - \hat{\mu}_*(t)|) \, \mathbb{1}\{A_t = *, Z_t\} \right).
\end{aligned}$$

Summing up the expectation value over all $t$ yields

$$\sum_{t=1}^{T} \mathbb{E}\left( \psi(|S - \hat{\mu}_*(t)|) \, \mathbb{1}\{A_t = *, Z_t\} \right)$$

$$\leqslant \sum_{t=1}^{T} \mathbb{E}\left( \psi(|S - \hat{\mu}_{*,t}|) \, \mathbb{1}\{\hat{\mu}_{*,t} \leqslant S\} \right)$$

$$= \sum_{n=1}^{T} \int_0^{\infty} \mathbb{P}\left( \psi(|S - \hat{\mu}_{*,n}|) \, \mathbb{1}\{\hat{\mu}_{*,n} \leqslant S\} \geqslant x \right) dx$$

$$= \sum_{n=1}^{T} \int_0^{\psi(0)} \mathbb{P}\left( \psi(|S - \hat{\mu}_{*,n}|) \, \mathbb{1}\{\hat{\mu}_{*,n} \leqslant S\} \geqslant x \right) dx$$

$$+ \sum_{n=1}^{T} \int_{\psi(0)}^{\infty} \mathbb{P}\left( \psi(|S - \hat{\mu}_{*,n}|) \, \mathbb{1}\{\hat{\mu}_{*,n} \leqslant S\} \geqslant x \right) dx$$

$$= \sum_{n=1}^{T} \int_0^{\psi(0)} \mathbb{P}\left( \hat{\mu}_{*,n} \leqslant S \right) dx + \sum_{n=1}^{T} \int_{\psi(0)}^{\psi(\infty)} \mathbb{P}\left( \hat{\mu}_{*,n} \leqslant S - \psi^{-1}(x) \right) dx,$$

noting that, since $\psi$ is increasing, for $x \leqslant \psi(0)$ the inequality $\psi(S - \hat{\mu}_{*,n})\mathbb{1}\{\hat{\mu}_{*,n} \leqslant S\} \geqslant x$ is equivalent to $\mathbb{1}\{\hat{\mu}_{*,n} \leqslant S\} = 1$, while for $x \geqslant \psi(0)$ it is equivalent to $\psi(S - \hat{\mu}_{*,n}) \geqslant x$. Further, note that if $x > \psi(\infty) := \lim_{y \to \infty} \psi(y)$ then the integrand is equal to 0.

We continue with the analysis of the same term and obtain

$$
\sum_{t=1}^{T} \mathbb{E}\big(\psi(|S - \hat{\mu}_*(t)|)\,\mathbb{1}\{A_t = *, Z_t\}\big)
$$

$$
\leqslant \sum_{n=1}^{T} \psi(0)\,\mathbb{P}\big(\hat{\mu}_{*,n} \leqslant S\big) + \sum_{n=1}^{T} \int_0^\infty \mathbb{P}\big(\hat{\mu}_{*,n} \leqslant S - u\big)\,\psi'(u)\,du
$$

$$
\leqslant \sum_{n=1}^{T} \psi(0)\exp\big(-\tfrac{n(\Delta_*^S)^2}{2}\big) + \sum_{n=1}^{T} \int_0^\infty \exp\big(-\tfrac{n(|\Delta_*^S|+u)^2}{2}\big)\,\psi'(u)\,du
$$

$$
\leqslant \frac{2\psi(0)}{(\Delta_*^S)^2} + \int_0^\infty \sum_{n=1}^{T} \exp\big(-\tfrac{n(|\Delta_*^S|+u)^2}{2}\big)\,\psi'(u)\,du
$$

$$
\leqslant \frac{2\psi(0)}{(\Delta_*^S)^2} + \int_0^\infty \frac{\psi'(u)}{e^{\frac{(|\Delta_*^S|+u)^2}{2}} - 1}\,du.
$$

Finally, by putting everything together, we obtain

$$
R_T^S = \sum_{i:\Delta_i^S > 0} \mathbb{E}(n_i(T))\,\Delta_i^S
$$

$$
\leqslant \sum_{i:\Delta_i^S > 0} \left(\Delta_i^S + \frac{8}{\Delta_i^S} + \frac{\Delta_i^S}{\psi\big(\frac{\Delta_i^S}{2}\big)}\left(\frac{2\psi(0)}{(\Delta_*^S)^2} + \int_0^\infty \frac{\psi'(x)}{e^{\frac{(|\Delta_*^S|+x)^2}{2}} - 1}\,dx\right)\right),
$$

which completes the proof. $\qquad\square$

As discussed by Bubeck et al. (2013) a simple choice for the potential function is $\psi(x) = x^2$, which gives a bound similar to that of Theorem 1. For refined choices for $\Psi$ we refer to Section 3 of (Bubeck et al., 2013). However, we note that in our setting one would need to know some of the gaps to design an algorithm that really gives improved bounds.

