# OpenReview forum: "Regret Bounds for Satisficing in Multi-Armed Bandit Problems"
_TMLR — Accepted by TMLR_

### Review · Reviewer_8YDu · 2023-06-23

**Summary Of Contributions:**

The paper analyzes a specific bandit setting in which the goal is to select an arm whose expected value is above a given threshold. The performances in this setting are evaluated using a different notion of regret than usual due to its characteristics. The authors propose two algorithms for the case in which the threshold is realizable and when not (i.e., when at least one arm has the expected value larger than the threshold). They show the regret bound for such algorithms. Finally, they propose a heuristic algorithm whose empirical performances are good in both realizable and non-realizable cases.

**Audience:**

Yes

**Broader Impact Concerns:**

I do not think such a theoretical work has a direct impact from an ethical point of view.

**Claims And Evidence:**

Yes

**Requested Changes:**

1) An overall check of the newly written parts would benefit the paper's readability.
2) "An experimental modification" is not clear to me. Please rephrase
3) In particular, this latter assumption of having..." I think that this comment is misleading. Bounded support implies subgaussianity.


**Strengths And Weaknesses:**

Most of the concerns I raised have been addressed in the new version of the paper.

The setting is clear. The algorithms are properly presented and analyzed.

The authors motivated the setting with a collaboration on the problem of transmission channel selection. In the new version of the paper, they put some effort into strengthening the motivation of the paper. However, I think adding some references and providing more details on the application would further prove the point of the necessity of this work.

The paper would require some rewriting since some concepts are sometimes not expressed linearly. Especially the newly added portion the writing is a bit too colloquial (e.g., "a lot of other").

---

### Review · Reviewer_A6jk · 2023-06-24

**Summary Of Contributions:**

This paper considers the multi-armed bandit problem with the satisficing objective. In this problem, the learner pursues an arm whose reward is no smaller than a given satisfaction level $S$, instead of the optimal arm. The authors give algorithms and analysis for both the realizable setting (where a satisficing arm exists), and the general setting (where a satisficing arm may not exist). With a newly defined notion of satisficing regret, the authors provide a constant satisficing regret bound when there is a satisficing arm, and recover the standard logarithmic regret bound when a satisficing arm does not exist. Experimental results demonstrate that the proposed algorithm outperforms standard algorithms not only in the satisficing setting, but also in the classic bandit setting.

**Audience:**

Yes

**Broader Impact Concerns:**

I believe this paper does not have negative ethical impacts.

**Claims And Evidence:**

Yes

**Requested Changes:**

Could the authors present the lower bound in a formal theorem form? This would improve the readability of this paper.

**Strengths And Weaknesses:**

**Strengths:**

1.	The satisficing objective in multi-armed bandit is interesting and well-motivated.
2.	This paper is well-written and easy to follow. The idea of seeking for satisficing arms is clear and well executed. The analysis and results match intuition.
3.	The authors design algorithms and provide regret bounds for the settings where the learner knows the existence of a satisficing arm and does not. When there is a satisficing arm, the proposed algorithm achieves a constant regret bound. When there is no satisficing arm, the proposed algorithm re-establishes the classic logarithmic regret bound.
4.	The authors also conduct empirical evaluations to show the performance superiority of their algorithm over existing algorithms in both the satisficing setting and classic bandit setting.
5. The authors' revision adresses my prior concerns on the intuition behind algorithms and related works.

**Weaknesses:**

It would be better to present the lower bound in a formal theorem form.

---

### Review · Reviewer_CM69 · 2023-06-25

**Summary Of Contributions:**

Please see the relevant section in my review for the previous version of the paper (Reviewer 1F3z).

**Audience:**

Yes

**Claims And Evidence:**

Yes

**Requested Changes:**

* Minor: it seems that contrarily to what is stated the figures still do not display error bars.


**Strengths And Weaknesses:**

I believe that the authors included in their revision the necessary changes for publication. In particular, I appreciate that
* previous work is now better stated and given enough credit, in my opinion. In particular, I appreciate that the authors acknowledged more clearly the inspiration from Bubeck et al. (2013) that solved a very similar problem. I also like the addition of thresholding bandits in the literature review.
* the presentation is clearer and focuses more on the insights (that may be the real interest of the paper) than on technical proofs (that are not novel, and now give enough credit to previous works). I appreciate that the proofs are still available but in a separate section, and are not presented as a contribution by the authors.
* Sat-UCB+ is clearly presented as an empirical method, and the authors focus more on explaining why it may be an interesting option to solve this problem on empirical. However, it is clear that the fact that Sat-UCB+ is not analyzed is still a weakness.

I still believe that the contribution of this work is relatively modest (see my previous review). In particular, with Sat-UCB+ not being analyzed the paper lacks a non-trivial theoretical contribution. However, the changes made to the paper make it interesting in the way it formalizes the problem, clearly present convincing answers, and fairly credit the existing works on similar topics.

In that sense, though not groundbreaking the paper may now be interesting for TMLR audience.

---

### Review · Reviewer_3JBt · 2023-06-27

**Summary Of Contributions:**

This study examines multi-armed bandit algorithms designed to minimize satisficing regret, which is defined as the deviation from a satisfaction level, in contrast to conventional regret based on cumulative rewards. The authors present algorithms that depend on whether the expected reward of the best arm exceeds the satisfaction level, and they establish upper bounds for their proposed algorithms. Finally, the authors validate the effectiveness of their proposed algorithms through simulation studies.


**Audience:**

Yes

**Broader Impact Concerns:**

Not applicable.

**Claims And Evidence:**

Yes

**Requested Changes:**

See above.

**Strengths And Weaknesses:**

I carefully read the revised draft and found that the authors effectively addressed the comments from the previous submission. I thank the authors for their revisions.

Overall, my primary comments are as follows:
- The problem setting seems to be novel.
- The algorithm proposed is well-suited for the problem.

As such, if the problem setting is indeed novel, I would not lean towards rejecting it. However, because the result seems somewhat trivial, I cannot strongly support acceptance.

Regarding the novelty, I have concerns about the novelty compared from the study of Reverdy (2017), which is one of the primary work cited in the draft. Is the problem different from theirs or not? Specifically, I have the following questions about the authors' claims:
- On page 2, the authors assert that "For $\delta = 0$, this coincides with our notion of satisficing regret as defined above." Is this indeed the case? While Reverdy (2017) considers that $1[\max\{\Delta^S_i, 0\} = 0]$ is a random variable, the authors do not. If $\delta = 0$, the regret in Reverdy (2017) becomes $R_ t = \Delta^S_{A_t}1[Pr[S_t = 1] < 1]$ (see (12) on page 3792 of Reverdy (2017)). If $S_t$ is deterministically $1$, the regret is zero because $Pr[S_t = 1]  = 1$; that is, $R_ t = \Delta^S_{A_t}0 = 0$ because $S_t = 1$ deterministically.
- On page 4, the authors compare their results with those in Reverdy (2017). They argue that they improved the regret bound. Although it appears improved, I am not fully convinced. I feel that it is not fair comparison because this study and Reverdy (2017) consider seemingly different problems.

For the added lower bounds, the authors mention the hypothesis testing and basic ideas for bandit lower bounds. However, both usually depend on the time length $T$ in some way, and it seems that they are irrelevant to this study's result. Anyway, if the authors mention lower bounds, I think it should be written in a more formal way.

In this study, it is seemingly important that $S$ is constant independent from $T$. If so, $S$ should be defined as such.

Minor comments:
- In the first line of page 1, the phrase "in general" is repeated. Is this a typo?
- In page 7, $\widehat{\mu}_{i, n}$, but it is not defined in the main body (although it appears in the appendix). If the authors use it in the main body, the definition also should be added there.
- Some claims (such as lower bounds) are provided without formal explanation. I believe that for readers, It is difficult to distinguish whether it is a well-known fact, conjecture or a new discovery by the author.

---

### Decision · Action_Editors · 2023-07-17

**Recommendation:** Accept as is

**Comment:**

This is a resubmission of paper #1035, which was originally rejected. The main reasons for the rejection were:

* Insufficient intuition behind algorithm designs and regret bounds.

* No lower bound. Therefore, the derived upper bounds may not be tight.

* Issues with experiments. For instance, the error bars overlapped and thus the statistical significance of some empirical results could not be judged.

These issues have been fixed. In addition, the authors positioned their contribution better with respect to prior works. Therefore, I suggest acceptance of this work.

**Audience:**

The general view of the reviewers is that this work is not groundbreaking and has weaknesses. In particular, the theory is rather straightforward and the best performing algorithm in experiments is not analyzed. Nevertheless, the presented setting is novel and I believe that it will appeal to the general bandit audience.

**Claims And Evidence:**

This is a resubmission of paper #1035, which was originally rejected. The main reasons for the rejection were the lack of clarity and that some claims were not supported by evidence. This was fixed.